# Spectrum of Neuroimaging Findings in Post-COVID-19 Vaccination: A Case Series and Review of Literature

**Shitiz Sriwastava** [1,2,3,4,*], **Ashish K. Shrestha** [5], **Syed Hassan Khalid** [1], **Mark A. Colantonio** [2], **Divine Nwafor** [2] **and Samiksha Srivastava** [4]

1  Department of Neurology, Rockefeller Neuroscience Institute, West Virginia University, Morgantown, WV 26506, USA; syedhassankhalid@gmail.com
2  School of Medicine, West Virginia University, Morgantown, WV 26506, USA; macolantonio@mix.wvu.edu (M.A.C.); dnwafor@mix.wvu.edu (D.N.)
3  West Virginia Clinical and Translational Science Institute, Morgantown, WV 26506, USA
4  Department of Neurology, Wayne State University, Detroit, MI 48201, USA; samiksha_sami@hotmail.com
5  Katmandu Medical College, School of Medicine, Kathmandu 44600, Nepal; sh_ashish@hotmail.com
*  Correspondence: shitiz.sriwastava@hsc.wvu.edu

**Abstract: Background and Purpose**: The novel severe acute respiratory syndrome coronavirus 2 (SARS-CoV-2) was first detected in Wuhan, China in December 2019. Symptoms range from mild flu-like symptoms to more severe presentations, including pneumonia, acute respiratory distress syndrome (ARDS), and even death. In response to the COVID-19 pandemic, the Emergency Use Authorization (EUA) approved the use of several vaccines. Because vaccines have been fast-tracked for emergency use, the short and long-term safety profile has been an area of concern. The aim of this paper is to extensively review published literature regarding post-COVID-19 vaccination neurological complications and characterize neuroimaging findings from three case presentations for early diagnosis and treatment. **Methods:** The analysis includes data from PubMed and Google Scholar. Articles included were retrieved from database inception beginning December 2020 with no language restrictions. Terms used include "SARS-CoV-2", "post Covid vaccination", "neurological complications", "Guillain-barre Syndrome", "Transverse-myelitis", "Cerebral Venous Sinus thrombosis", and "Cerebral hemorrhage". **Results:** The literature review yielded several neurological complications post vaccination, including cerebral sinus venous thrombosis, transverse myelitis, Guillain-Barré Syndrome and optic neuritis, to name a few. Patient case presentation findings were consistent with documented results in published literature. **Conclusions:** We present a case series with a thorough literature review documenting adverse neurological affects following COVID-19 vaccination. Our case presentations and literature review highlight the importance of neuroimaging when diagnosing post-COVID-19 vaccination adverse effects. MRI imaging study is an important tool to be considered in patients presenting with post-COVID-19 vaccination-related unexplained neurological symptoms for accurate diagnosis.

**Keywords:** COVID-19 vaccination; GBS; transverse myelitis; optic neuropathy

## 1. Introduction

Coronavirus disease 19 (COVID-19) is a leading cause of death globally and continues to detrimentally impact healthcare and economic sectors. In response to the COVID-19 pandemic, several vaccine initiatives were rapidly devised to minimize morbidity and mortality associated with SARS-CoV-2, with an end goal of eventually reaching herd immunity. Recently published data from a number of clinical studies suggest these vaccines are effective against preventing severe COVID-19 disease irrespective of design or strategy used in vaccine development [1]. Thus, emergency use authorization and vaccination campaigns were rapidly initiated to curb the spread of SARS-CoV-2, and to our knowledge, it is estimated that a total of 5.46 billion doses of vaccine have been administered worldwide [2].

The global vaccination campaign has been effective in reducing morbidity and mortality of COVID-19 in vaccinated individuals; however, several adverse effects following immunization (AEFI) have been noted. Typical common side effects following vaccination include pain, swelling, localized erythema over the injection site, fever, chills, fatigue, myalgia, muscle pain, vomiting, arthralgia, and lymphadenopathy [3–6]. Mild neurological symptoms including headache, dizziness, myalgia, muscle spasms, and paresthesia have also been reported. A small number of case reports have demonstrated serious post-vaccination neurological sequelae ranging from generalized seizures, Guillain Barre Syndrome (GBS), and transverse myelitis [7]. Other neurological manifestations, such as facial nerve palsy, acute disseminated encephalomyelitis, and stroke have been reported in the Vaccine Adverse Event Reporting System (VAERS) of the Centers for Disease Control (CDC) related to COVID-19 Pfizer-BioNTech, Moderna and Johnson & Johnson's COVID-19 vaccines [8].

Here in, we report a case series of transverse myelitis, chronic inflammatory demyelinating polyneuropathy and optic neuritis following use of Moderna, Johnson & Johnson's and Pfizer-BioNTech COVID-19 vaccination, respectively. Furthermore, we briefly provide a review of the literature on clinical management and the relevance of neuroimaging in patients presenting with post-COVID vaccination neurological sequelae.

## 2. Methods

The analysis includes data from PubMed and Google Scholar. Articles included were retrieved from database inception beginning December 2020 with no language restrictions. Terms used include "SARS-CoV-2", "post Covid vaccination", "neurological complications", "Guillain-barre Syndrome", "Transverse-myelitis", "Cerebral Venous Sinus thrombosis" and "Cerebral hemorrhage". Inclusion criteria were based on patients aged 18 and older with reported post-COVID-19 vaccination-related neurological complications with relevant imaging findings. Patients were analyzed based on the vaccination they received, CNS or PNS disorders they presented with, imaging findings and treatment they received. A cutoff date of August 30, 2021 was used. We only included Pfizer-BioNTech BNT162b2, Moderna mRNA-1273, Johnson & Johnson's Janssen (J&J/Janssen), and AstraZeneca vaccination-related neurological complications with associated neuroimaging findings. The exclusion criteria for the studies include: (1) Patient age < 18 years; (2) duplicate articles which involved repetition of cases; (3) articles in languages other than English; (4) studies that had no available individual patient's data; (5) editorials (Table 1).

## 3. Case Summary

A 67-year-old woman—1-day post-Moderna COVID-19 vaccination—experienced lower extremity weakness. Five days post-vaccination, the patient presented to the emergency department (ED) with difficulties ambulating. On examination, muscle strength was grade 4/5 in the lower extremity and 5/5 in the upper extremity. Reflexes were brisk at the knee. Magnetic resonance imaging (MRI) of the cervical spine revealed intramedullary cord signal changes extending from C1-C3 with patchy enhancement on post-contrast imaging (Figure 1). Brain MRI revealed nonspecific deep white matter changes. Thoracic spine MRI was unremarkable. Cerebrospinal fluid (CSF) yielded normal chemistry, cell count, and absent oligoclonal bands. Serum neuromyelitis optica (NMO) and myelin oligodendrocyte glycoprotein (MOG) were negative. The patient responded well to IVIG and plasmapheresis, with partial recovery of lower extremity motor strength. At follow-up, the patient was able to stand under her own power and ambulate with assistance.

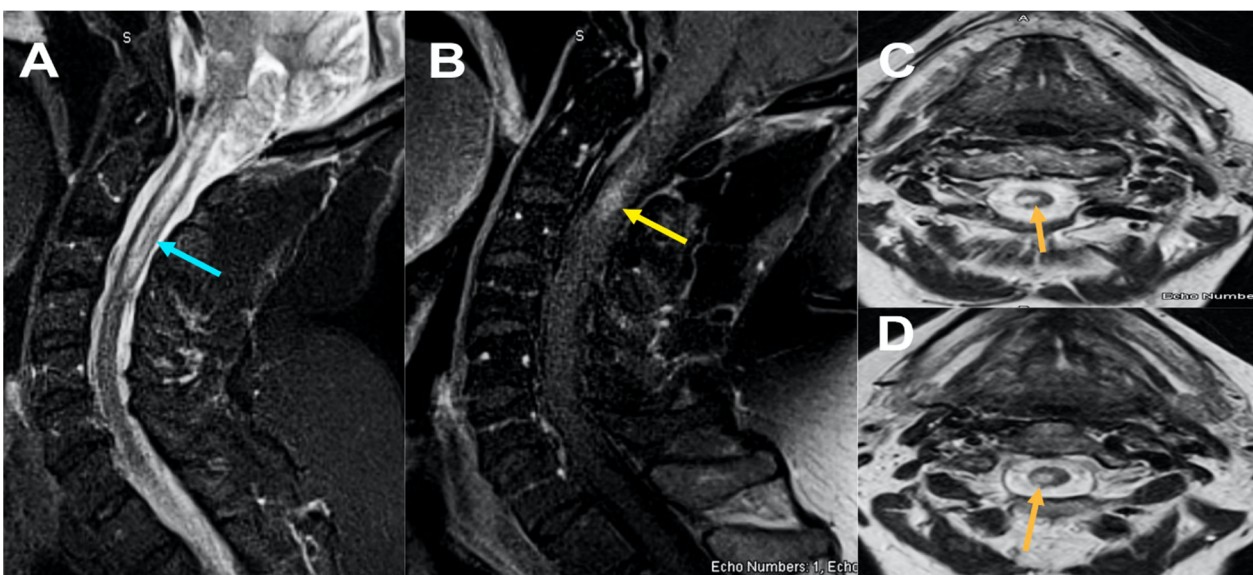

**Figure 1.** MRI Cervical spine Figure 1, sagittal STIR (short tau inversion recovery) images (**A**); sagittal T1 post contrast (**B**) and axial T2 weighted images at C2 and C3 level (**C**) and (**D**) reveals ill-defined long segment signal alteration with mild cord expansion extending from C1-C3 level (yellow arrow) (**A**) with corresponding hyperintensity on axial images (**C**) and (**D**) and abnormal enhancement on post contrast at C2 level (**B**) (yellow arrow).

A 41-year-old male presented to the ED post-COVID-19 vaccination (Johnson & Johnson vaccine) with complaints of generalized weakness, difficulty ambulating two weeks post-vaccination. On physical examination MRC grade 4/5 in lower extremity throughout. Reflexes were absent. Upon admission, MRI of the brain, lumbar spine, and cervical spine without contrast were unremarkable. CSF revealed increased protein, normal cell count, and albumino cytological dissociation. EMG/NCS revealed findings consistent with demyelinating polyneuropathy. The patient was treated with and responded well to IVIG. At a two-month follow-up, the patient stated he was ambulating with assistance. Follow-up lumbosacral spinal MRI imaging with/without contrast revealed thickened cauda equina nerves. These findings raised concern for disease progression to chronic inflammatory demyelinating polyneuropathy (CIDP) (Figure 2).

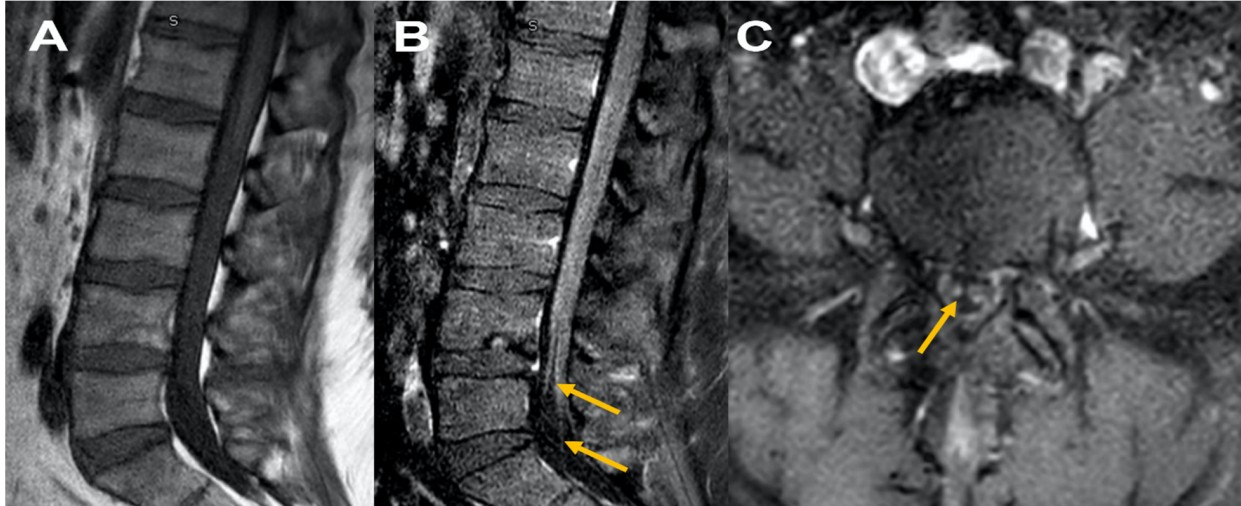

**Figure 2.** MRI Lumbar spine Figure 2, sagittal T1 plain (**A**) and corresponding sagittal post contrast T1 (**B**) and axial cuts at L5 level (**C**) reveal normal appearing disc and facets, Figure 1A; cauda equina nerve roots are thickened with no abnormal enhancement consistent with chronic inflammatory demyelinating polyneuropathy (**B**) and (**C**).

A 42-year-old male presented to the ED with complaints of blurry vision and 3-day left eye pain worsened with movement. Upon examination, diffuse gray field vision in the left eye was noted. The patient received both doses of the Pfizer-BioNTech COVID-19 vaccination 3 months ago. Cranial nerve 3, 4, and 6 examination revealed horizontal diplopia in the left eye. Muscle strength 5/5 bilaterally and reflexes 2+ bilaterally. Further eye examination revealed a positive relative afferent pupillary defect in the left eye with clear disc margins. Orbital MRI revealed left optic nerve enhancement (Figure 3). MRI brain unremarkable. CSF was significant for increased protein. Oligoclonal bands were absent and NMO, MOG antibodies were negative. The patient showed significant improvement at discharge with IV solumedrol and oral prednisone treatment.

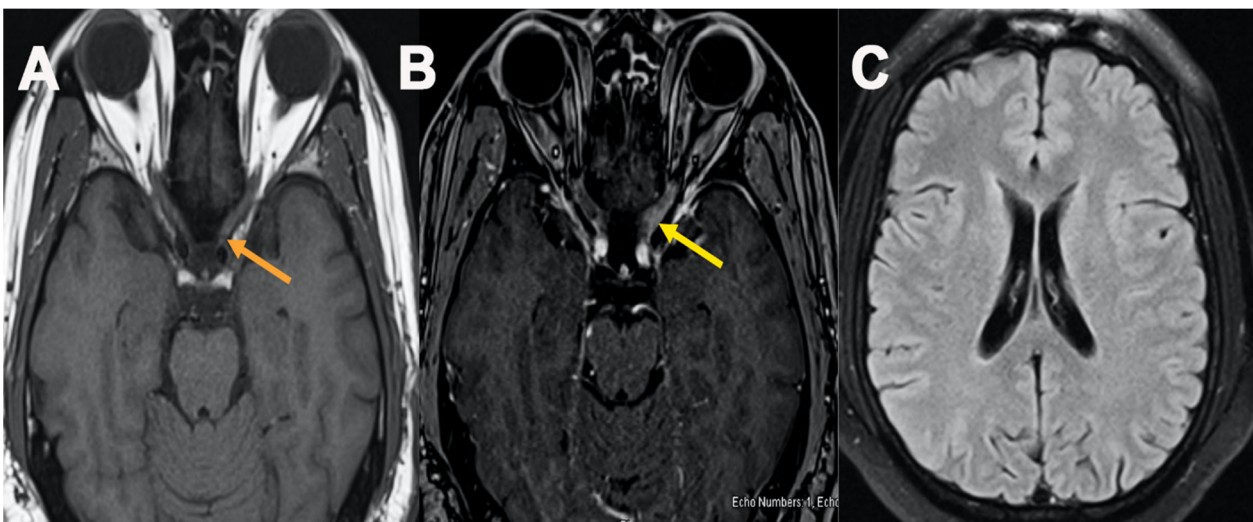

**Figure 3.** MRI orbit Figure 3, axial T1 plain (**A**) and post contrast fat saturation axial images (**B**) showed abnormal enhancement within the intracanalicular portion of left optic nerve (**B**) (yellow arrow), MRI brain axial FLAIR cut at level of lateral ventricle showed no abnormal T2/FLAIR signal changes (**C**).

## 4. Discussion

Among the 3 vaccines approved by the Emergency Use Authorization (EUA), Pfizer-BioNTech BNT162b and Moderna mRNA-1273 are mRNA vaccines that code for S-protein antigens. Ultimately, these vaccines are composed of SARS-CoV-2 glycoproteins and lipid nanoparticles adjuvants, which induce B-cell and T-cell response [9]. The Janssen COVID-19 vaccine is a recombinant, adenovirus vector-based vaccine that encodes the SARS-CoV-2 spike protein. The adenoviruses vector enters cells and uses the host machinery to translocate spike proteins to the cellular surface, enabling recognition by the host immune system. This triggers the production of antibodies against COVID-19 antigens [10].

Adverse reactions can occur with the administration of any vaccine. Several neurological manifestations following the COVID-19 vaccination have been reported. The National Board of UK and Vaccine Adverse Event Reporting System (VAERS), which is managed by the CDC and FDA, has reported transverse myelitis, GBS, Bell' palsy, cerebral venous sinus thrombosis, and stroke following administration of Pfizer-BioNTech BNT162b, Moderna mRNA-1273 Oxford-AstraZeneca and Janssen (Johnson & Johnson) vaccine [8,11–13] (Tables 2 and 3). Complications and treatment outcomes of complications were further explored.

Table 1 describes demographics, neurological features, MRI and CT scan findings, and treatment outcomes of neurological complications following COVID-19 vaccination. According to the literature, 29 cases affecting the central and peripheral nervous system have been reported. Among them, 11 cases involved the peripheral nervous system (PNS), with GBS [8] being the predominant case, followed by Parsonage-Turner syndrome (*n* = 2) and Phantosmia (*n* = 1). The other 18 cases involved central nervous system (CNS) mani-

festations, with cerebral sinus venous thrombosis (CVST) (*n* = 10) being the most common complication, followed by transverse myelitis (*n* = 3) and individual cases of multiple sclerosis, ischemic stroke, hemorrhagic stroke, superior ophthalmic vein thrombosis and ADEM.

Among the eight cases of reported GBS, thickening and enhancement of the cauda equina nerve roots was the most common MRI lumbar spine finding observed in four cases [14–17], while three cases of facial diplegia variant of GBS were also reported [18–20]. Imaging revealed enhancement of the facial nerve at the fundus of the internal acoustic meatus. There are two cases of Parsonage-Turner syndrome were reported with MR neurography findings were significant for hyperintensity of the median and suprascapular nerve [21]. Phantosmia was a complication described by Keir et al., which yielded hyperintensity and enhancement of the olfactory bulb and tract on imaging [22].

Among CNS manifestations the most common MR or CT venography CNS findings were superior sagittal sinus thrombosis (*n* = 8), followed by thrombosis of the transverse and sigmoid sinus (*n* = 3) [23–28]. There was a reported case with MRI findings significant for superior ophthalmic vein thrombosis (SOVT) [29]. A case of possible CNS demyelinating disorder was also reported and yielded multiple partially confluent lesions in periventricular, cortical, juxtacortical, splenium of corpus callosum, and infratentorial region with spatial dissemination along with contrast-enhancing lesion at the T-6 level [30]. Similarly, neuroimaging revealed frontal lobe hyperintensities in a case of ADEM described by Vogrig et al. [31] (Table 1).

Among the GBS cases, CSF findings demonstrated a high protein level (>45 mg/dL) in all cases with normal WBC (<5 cell/mm$^3$) in four cases. Among the CNS cases, CSF analysis was performed only in cases of transverse myelitis, multiple sclerosis, and ADEM (*n* = 5). Elevated protein (>45 mg/dL) was observed in two cases and pleocytosis was observed (>5 cell/m$^3$) in three cases (Table 1).

Treatment varied between cases involving the CNS and PNS. IVIG and gabapentin were chosen as treatment modalities for PNS cases. All cases recovered without mortality. However, seven fatalities were recorded in the CNS cases, all of which were mostly attributed to brainstem death (Table 1).

Cerebral venous sinus thrombosis (CVST) has been observed as a complication post-COVID 19 vaccination. CVST occurs due to incomplete or complete occlusion of the draining veins of the cerebral venous system, resulting in decreased cerebrospinal fluid absorption and subsequent backflow and hypertension [23]. Literature analysis revealed patients presented with thrombotic complications and concomitant thrombocytopenia following administration of the ChAdOx1 vaccination. Presentation of prothrombotic features with moderate-to-severe thrombocytopenia after 1–2 weeks ChAdOx1 vaccination displayed clinical similarities to heparin-induced thrombocytopenia; however, these patients had no history of heparin administration [32]. It is known that adenovirus causes activation of platelet aggregation, and interaction of free DNA in the vaccine can trigger the production of PF4-reactive antibodies associated with ChAdOx1 vaccination. mRNA vaccines may cause an immunological response leading to surface receptor dysregulation or activate thrombosis formation. This phenomenon is known as vaccine-induced thrombotic thrombocytopenia (VITT) and has shown devastating effects in otherwise healthy young adults [33,34].

Physicians should be vigilant for patients presenting with arterial and venous thrombosis 1–3 weeks post-vaccination. Suspected cases should be investigated with ELISA to detect PF4 antibodies to confirm a diagnosis of VITT. Treatment decisions should not be delayed and must be initiated by starting anticoagulation and intravenous immunoglobulin [32].

Our cases series presentations revealed findings consistent with those presented in the published literature. Here, we discuss the disease process associated with our cases and the rationale for diagnosis. Transverse myelitis is a focal inflammatory neuro-immune spinal cord disorder. Several complications have been observed post-COVID

vaccination. Symptoms include rapid onset motor weakness, sensory alterations, and autonomic dysfunction. Etiology may be infectious, para-infectious, systemic autoimmune, or ischemic diseases [35,36]. Although rare, transverse myelitis has been observed with measles, mumps and rubella, influenza, oral polio, and hepatitis B vaccination [37–40]. After the administration of the vaccine, an immune response is induced. On rare occasions, molecular mimicry may occur when our body cannot distinguish between host or enemy, resulting in autoimmunity and destruction of host cells. Transverse myelitis may also be due to the acceleration and overaction of prior autoimmune processes induced by foreign antigens [41,42].

Our case fulfills diagnostic criteria for transverse myelitis according to the Transverse Myelitis Consortium Working Group. Clinical presentation includes motor and sensory abnormalities, along with urinary retention [43,44]. This is consistent with our case findings. Multiple sclerosis (MS) was ruled out due to the absence of multiple sclerosis-like lesions in the brain, described as dissemination in time and space. Oligoclonal bands were absent in CSF, also diagnostic of MS [29,45]. After a prompt diagnosis by neuroimaging, the patient recovered with corticosteroid and PLEX therapy. Similar improvements were seen in cases by Pagenkopf, Malhotra, and Fitzsimmons et al. [44,46,47].

Our second case presented with clinical findings consistent of CIDP following COVID-19 vaccination. CIDP is an inflammatory polyradiculoneuropathy seen after viral infections or vaccinations. Pfizer and Moderna vaccines, constructed as messenger RNA vaccines, may trigger an autoimmune response, leading to the formation of antibodies against the myelin sheath [14]. AstraZeneca (ChAdOx1) and the Johnson and Johnson vaccine contain the Simian adenovirus as a vector, which induces antibodies against the spike glycoprotein of SARS-CoV-2 for immune system recognition [48,49]. Waheed et al. were the first to report a case of GBS following COVID-19 vaccination with the Pfizer vaccine. In this literature review, the patients' demyelinating polyneuropathy including GBS cases made a partial or full recovery with the use of IVIG, steroids, and supportive care (Table 1).

Our last patient presented with clinical symptoms consistent with acute optic neuritis (ON). Patients present with unilateral painful loss of vision due to demyelination. Optic nerve lesions, papillitis, myelin oligodendrocyte glycoprotein (MOG) positive antibodies, and perineuritis are consistent with ON [50,51]. Based on literature review, vaccine induced antibodies can cross react with proteins present in the retinal pigment epithelial cells, optic nerve myelin, and antigens in large arteries leading to optic neuritis, anterior ischemic optic neuropathy (AAION), and bilateral acute zonal occult outer retinopathy (AZOOR) [52].

Alvarez et al. published a global report of 73 cases based on the frequency of bilateral presentation, age distribution, and radiological features more commonly found in immune-mediated ON. Most events (67%) occurred after vaccination with AstraZeneca, followed by Pfizer-BioNTech (26%) and Sinovac (7%). The author determined the diagnosis on enhancement, swelling, adnexal involvement, and intracranial pathology, especially of inflammatory/demyelinating lesions seen on post-gadolinium MRI sequences and optic nerve lesion lengths on STIR (short tau inversion recovery). The result reported no statistical difference in severity, treatment, and outcome when compared to Pfizer, AstraZeneca, and Sinovac vaccine [51]. Our goal is to highlight and recognize Post-SARS-CoV-2 vaccination ON as an extremely rare adverse plausible event with a generally good outcome.

**Table 1.** Study origin, demographics, CSF, MRI findings, and outcomes in COVID-19 vaccination and CNS inflammatory disorders.

| Author/ Country | Patient Age/Gender | Vaccination | Time Duration from COVID-19 Vaccination to Neurological Symptom Onset | Diagnosis | Neurological Presentation | CSF Findings Cell Count, Protein, Glucose, Oligoclonal Bands | MRI Brain/Spine Finding | CT Finding | CTA/MRA Finding | CT/MR VENOG-RAPHY Findings | Treatment | Outcome |
|---|---|---|---|---|---|---|---|---|---|---|---|---|
| Waheed et al./US [14] | 82 years/F | Pfizer | 14 days | GBS | Generalized malaise, body aches and difficulty walking | Showed albumino-cytologic dissociation Protein of 88 mg/dL, Cell Count: WBC of 4/mm$^3$ | MRI L spine: Enhancement of cauda equina nerve roots | Normal | N/A | N/A | IVIG | Recovered |
| Malhotra et al./India [46] | 36 years/M | AstraZeneca | 8 days | Transverse Myelitis | Abnormal sensation in the lower limbs. | Protein: 54 mg/dL Cell count: normal Oligoclonal bands negative | MRI spine: T2-hyperintense lesion at C6-C7 with enhancement | N/A | N/A | N/A | IV steroids | Recovered |
| Keir et al./US [22] | 57 years/F | Pfizer | Post-second dose | Phantosmia | Smelling smoke, hyposmia and headaches | N/A | MRI brain: enhancement of the left greater than right olfactory bulb and bilateral olfactory tracts and hyperintensity in olfactory bulbs and tracts | Normal | CTA no vessel occlusion or aneurysm | N/A | None | Recovered |
| Razok et al./Qatar [15] | 73 years/M | Pfizer | 20 days | GBS | Progressive lower limb weakness | Protein: 80 mg/dL (elevated) Cell Count: Normal Glucose: Normal Oligoclonal band: negative | MRI L spine: bilateral nerve root enhancement in the lumbar region and the upper part of the cauda equina | Normal | N/A | N/A | IVIG | Recovered |
| Jose et al./India [48] | 66 years/M | AstraZeneca | 12 days | GBS | Sensorimotor weakness Proximal lower limb with mild hand grip weakness | Cell protein: 84 mg/dL (elevated) Cell count: Normal Glucose: Normal | MRI L Spine: normal | N/A | N/A | N/A | IVIG and Steroids | Partially recovered |
| Prasad et al./US [20] | 41 years/M | Janssen | 21 days | GBS/BFP variant | Difficulty feeding and ambulating | Cell count: 50/mm$^3$ Protein: 562 mg/dL Glucose: 67 mg/dL | MRI L spine with contrast showed thickening of cauda equina nerve roots | N/A | N/A | N/A | IVIG | Recovered |

**Table 1.** *Cont.*

| Author/ Country | Patient Age/Gender | Vaccination | Time Duration from COVID-19 Vaccination to Neurological Symptom Onset | Diagnosis | Neurological Presentation | CSF Findings Cell Count, Protein, Glucose, Oligoclonal Bands | MRI Brain/Spine Finding | CT Finding | CTA/MRA Finding | CT/MR VENOG-RAPHY Findings | Treatment | Outcome |
|---|---|---|---|---|---|---|---|---|---|---|---|---|
| Márquez Loza AM et al./US [16] | 60 years/M | Janssen | 10 days | GBS | Pain in her back and leg. Nausea, vomiting, headache and diplopia | CSF Protein: 140 mg/dL, Cell count: 9 nucleated cells/mm$^3$ Glucose: Normal | MRI L spine: demonstrated enhancement of the cauda equina | N/A | N/A | N/A | IVIG | Recovered |
| Queler SC et al./US [21] | 49 years/M | Pfizer | 13 hr post-first dose | Parsonage-Turner Syndrome | Severe, electric shooting pain in his left volar forearm | N/A | MR neurography: hyperintensity of the anteromedially positioned fascicular bundle of the median nerve were detected | N/A | N/A | N/A | Gabapentin and oral steroids | Partial Recovery with weakness |
| Queler SC et al./US [21] | 44 years/M | Moderna | 18 days | Parsonage-Turner Syndrome | Sudden onset, intense, cramping pain in the left lateral deltoid region | N/A | MR Neurography: hyperintensity and multiple, focal, hourglass-like constrictions of the suprascapular nerve with edema | N/A | N/A | N/A | Gabapentin | Improved |
| Havla et al./Germany [30] | 28 years/F | Pfizer | 6 days | Multiple sclerosis | Left abdominal neuropathic pain, sensory impairment below the T6 level | CSF cell count: 7/mm$^3$ Oligoclonal band: Positive | MRI brain: Multiple confluent T2/FLAIR lesions in periventricular, cortical, juxtacortical, splenium of corpus callosum and infratentorial region no enhancement. MRI spine: contrast-enhancing lesion at the T6 level | N/A | N/A | N/A | IV steroids and plasma-pheresis | Improved |

**Table 1.** *Cont.*

| Author/ Country | Patient Age/Gender | Vaccination | Time Duration from COVID-19 Vaccination to Neurological Symptom Onset | Diagnosis | Neurological Presentation | CSF Findings Cell Count, Protein, Glucose, Oligoclonal Bands | MRI Brain/Spine Finding | CT Finding | CTA/MRA Finding | CT/MR VENOG-RAPHY Findings | Treatment | Outcome |
|---|---|---|---|---|---|---|---|---|---|---|---|---|
| Patel et al./UK [17] | 37 years/M | AstraZeneca | 14 days | GBS | Persistent back pain, distal paraesthesia in hands and feet, symmetrical progressive ascending muscle weakness | CSF protein:177 mg/dL Glucose: 70 mg/dL Cell Count: <1/mm$^3$ | MRI L spine: illustrated globally thickened cauda equina nerve root particularly at the level of S1 | N/A | N/A | N/A | IVIG | Improved |
| Azam et al./UK [18] | 67 years/M | AstraZeneca | 15 days | GBS BFP variant | progressive worsening of the gait, bilateral leg, bilateral facial weakness, and difficulty in chewing food | CSF Protein: 390 mg/dL Glucose: 86 mg/dL | MRI brain: showed enhancement of the facial nerve bilaterally at the fundus of the internal auditory meatus extending into the labyearsinthine segment | Normal | N/A | N/A | IVIG | Improved |
| Fitzsimmons et al./US [47] | 63 years/M | Moderna | 2 days | Transver-semyelitis | Shooting pain and numbness in lower legs and buttocks. Difficulty urinating and constipation | CSF: glucose 74 mg/dL CSF total protein 37 mg/dL Cell count: total nucleated cell counts 3/mm$^3$ | MRI brain: Few punctate hyperintensities in bilateral corona radiata. MRI spine: Increased T2 cord signal distal spinal cord and conus with questionable associated enhancement | N/A | N/A | N/A | Oral and IV steroids | Recovered |

**Table 1.** *Cont.*

| Author/ Country | Patient Age/Gender | Vaccination | Time Duration from COVID-19 Vaccination to Neurological Symptom Onset | Diagnosis | Neurological Presentation | CSF Findings Cell Count, Protein, Glucose, Oligoclonal Bands | MRI Brain/Spine Finding | CT Finding | CTA/MRA Finding | CT/MR VENOG-RAPHY Findings | Treatment | Outcome |
|---|---|---|---|---|---|---|---|---|---|---|---|---|
| Schultz et al./Norway [33] | 54 years/F | AstraZeneca | 7 days | CVST | Hemiparesis on the left side of her body | N/A | N/A | Right frontal hemor-rhage | N/A | A CT scan with venogra-phy showed a massive cerebral vein thrombo-sis with global edema and growth of hematoma | Oral steroids, IVIG, Heparin, decom-pressive Hemi-craniec-tomy | Deceased |
| Vogrig et al/Italy [31] | 56 years/F | Pfizer | 14 days | ADEM | Left side unsteady gait and clumsiness of left arm. | CSF cell count: pleocytosis (80 cells/mm$^3$) CSF protein: Normal Glucose: Normal | MRI brain hyperintensity involving the frontal white matter, with the largest lesion on the left side | N/A | N/A | N/A | Oral Steroids | Recovered |
| Dutta et al./India [23] | 51 years/ M | AstraZeneca | 5 days after first dose | CVST | Holocephalic headache, vomiting | N/A | N/A | N/A | N/A | MR Venogra-phy:Thrombosis in superior sagittal sinus and transverse sinus with presence of extensive venous collaterals | LMWH | Improved |

**Table 1.** *Cont.*

| Author/ Country | Patient Age/Gender | Vaccination | Time Duration from COVID-19 Vaccination to Neurological Symptom Onset | Diagnosis | Neurological Presentation | CSF Findings Cell Count, Protein, Glucose, Oligoclonal Bands | MRI Brain/Spine Finding | CT Finding | CTA/MRA Finding | CT/MR VENOG-RAPHY Findings | Treatment | Outcome |
|---|---|---|---|---|---|---|---|---|---|---|---|---|
| Rossetti et al./US [19] | 38 years/M | Janssen | 14 weeks | GBS/BFP | Bilateral hand and foot paresthesias, dysarthria, bilateral facial weakness | CSF Glucose: 73 mg/dL CSF protein: 181 mg/dL Cell Count: 7/mm$^3$ | MRI brain: bilateral internal auditory canal fundi which carry CN-VII and CN-VIII | N/A | N/A | N/A | IVIG | Improved |
| Bayas et al./Germany [29] | 55 years/F | AstraZeneca | 10 days | SOVT | Conjunctival congestion, retro-orbital pain, and diplopia | N/A | MRI brain: Showed superior ophthalmic vein thrombosis (SOVT) | N/A | N/A | N/A | Heparin | Recovered |
| Blauenfeldt et al./Denmark [53] | 60 years/F | AstraZeneca | 7 days | Ischemic stroke | Persistent abdominal pain. Left-sided weakness and eye deviation to the right | N/A | MRI brain: Diffusion restriction and infarction in the entire area supplied by the right middle cerebral artery | Midline shift of 12 mm | N/A | N/A | Dalteparin, hemi-craniec-tomy | Deceased |
| Castelli et al./Italy [24] | 50 years/M | AstraZeneca | 11 days after First Dose | CVST | Severe headache, loss of strength in the right lower limb, unstable walking and slight visual impairment | N/A | N/A | Intraparen-chymal hemor-rhage in the left hemi-sphere. | CTA multiple bleeding within the parenchy-mal and left transverse and sigmoid sinuses, thrombosis | N/A | Bilateral decom-pressive craniec-tomy | Deceased |

**Table 1.** *Cont.*

| Author/ Country | Patient Age/Gender | Vaccination | Time Duration from COVID-19 Vaccination to Neurological Symptom Onset | Diagnosis | Neurological Presentation | CSF Findings Cell Count, Protein, Glucose, Oligoclonal Bands | MRI Brain/Spine Finding | CT Finding | CTA/MRA Finding | CT/MR VENOG-RAPHY Findings | Treatment | Outcome |
|---|---|---|---|---|---|---|---|---|---|---|---|---|
| Agostino et al./Italy [25] | 54 years/F | AstraZeneca | 12 days | CVST | Acute cerebrovascular accident | N/A | MRI brain: restricted diffusion in pons, mesencephalon, the right superior cerebellar hemisphere with the vermis and the right posterior temporal lobe | Presence of multiple suba-cute intra-axial hemor-rhages | CTA partial thrombosis of the vein of Galen; MRA: acute basilar thrombosis associated with superior coronal and sagittal sinus thrombosis. | N/A | No | Deceased |
| Mehta et al./UK [26] | 32 years/M | AstraZeneca | 9 days post first dose | CVST | Thunderclap headache and subsequent left-sided in-coordination and hemiparesis | N/A | N/A | Clot ex-panding the middle to anterior third of the superior sagittal sinus, seen as an area of hyper density | N/A | Superior sagittal sinus and cortical vein thrombosis and cortical oedema with areas of parenchy-mal and subarach-noid hemor-rhage | No | Deceased |

**Table 1.** *Cont.*

| Author/ Country | Patient Age/Gender | Vaccination | Time Duration from COVID-19 Vaccination to Neurological Symptom Onset | Diagnosis | Neurological Presentation | CSF Findings Cell Count, Protein, Glucose, Oligoclonal Bands | MRI Brain/Spine Finding | CT Finding | CTA/MRA Finding | CT/MR VENOG-RAPHY Findings | Treatment | Outcome |
|---|---|---|---|---|---|---|---|---|---|---|---|---|
| Mehta et al./UK [26] | 25 years/M | AstraZeneca | 6 days | CVST | Meningitis headache, photophobia, vomiting, petechial rash, gum bleeding, left hemiparesis, left hemisensory loss | N/A | N/A | Large volume clot within the superior sagittal sinus | N/A | Large filling defect in the anterior two thirds of the superior sagittal sinus | Unfractio-nated heparin | Deceased |
| Wolf et al./Germany [27] | 22 years/F | AstraZeneca | 4 days | CVST | Frontally accentuated headaches, self-limited generalized epileptic seizures | N/A | MRI brain: Blood in the subarachnoid space adjacent to the falx cerebri bilateral. The superior sagittal sinus, the left transverse sinus, and the sigmoid sinus were thrombosed | N/A | MRA: Revealed thrombotic occlusion of the superior sagittal sinus | N/A | Enoxaparin | Recovered |
| Wolf et al./Germany [27] | 46 years/F | AstraZeneca | 8 days | CVST | Severe headaches, focal neurologic symptoms with mild aphasia and hemianopia to the right | N/A | MRI brain: Thrombotic occlusion of the superior sagittal sinus and the left transverse sinus and sigmoid sinus. | N/A | Occlusion of the superior sagittal sinus and the left transverse sinus and the sigmoid sinus | N/A | Enoxaparin, Dabiga-tran, Endovas-cular recanal-ization | Recovered |

**Table 1.** *Cont.*

| Author/ Country | Patient Age/Gender | Vaccination | Time Duration from COVID-19 Vaccination to Neurological Symptom Onset | Diagnosis | Neurological Presentation | CSF Findings Cell Count, Protein, Glucose, Oligoclonal Bands | MRI Brain/Spine Finding | CT Finding | CTA/MRA Finding | CT/MR VENOG-RAPHY Findings | Treatment | Outcome |
|---|---|---|---|---|---|---|---|---|---|---|---|---|
| Wolf et al./Germany [27] | 36 years/F | AstraZeneca | 7 days | CVST | Severe headaches | N/A | MRI Brain showed a thrombotic occlusion of the straight sinus and a non-occlusive thrombus in the superior sagittal sinus. | N/A | occlusion of the straight sinus and a non-occlusive thrombus of the superior sagittal sinus. | N/A | Enoxaparin, Dabiga-tran, Endovas-cular recanal-ization | Recovered |
| Bjornstad-Tweng et al./Norway [54] | 30 years/F | AstraZeneca | 7 days | Cerebral hemorrhage | Severe headaches, slurred speech, unco-ordinated movements and reduced consciousness | N/A | N/A | Showed right-sided hemor-rhage and incipient hernia-tion. | CT angiog-raphy showed no evidence of an aneurysm. | N/A | Tranexamic acid | Deceased |
| Pagenkopf et al./Germany [44] | 45 years/M | AstraZeneca | 8 days | Transverse myelitis | Headache, thoracic back pain and general weakness, urinary retention; acute flaccid tetraparesis, emphasizing lower limbs, and a sensory level at Th9 | CSF Cell Count: pleocytosis of 481 cells/$\mu$L Protein: (140 mg/dL), Glucose: normal Negative oligoclonal bands. | MRI brain: Normal MRI Spine: hyperintense signal of the spinal cord with wide axial and longitudinal extent reaching from C3 to Th2 without gadolinium enhancement | N/A | N/A | N/A | IV corticos-teroids | Improved |

**Table 1.** *Cont.*

| Author/ Country | Patient Age/Gender | Vaccination | Time Duration from COVID-19 Vaccination to Neurological Symptom Onset | Diagnosis | Neurological Presentation | CSF Findings Cell Count, Protein, Glucose, Oligoclonal Bands | MRI Brain/Spine Finding | CT Finding | CTA/MRA Finding | CT/MR VENOG-RAPHY Findings | Treatment | Outcome |
|---|---|---|---|---|---|---|---|---|---|---|---|---|
| Zakaria Z et al./Malaysia [28] | 49 years/M | Pfizer | 16 days | CVST | Moderate headache and dizziness | N/A | N/A | Showed cordlike hyper-attenuation within the left transverse and sigmoid sinus suggestive of cord or dense clot sign | N/A | Long segment-filling defect and empty delta sign within the superior sagittal sinus (SSS), extending in left transverse sinus, and sigmoid sinus to proximal internal jugular vein | Apixaban, Clopidogrel | Recovered |

Abbreviation: MRI: Magnetic resonance imaging, FLAIR: Fluid-attenuated inversion recovery, CSF: Cerebrospinal fluid, CTA: Computed tomography angiography, GBS: Guillain-Barré syndrome, IVIG: Intravenous immunoglobulin, CVST: Cerebral venous sinus thrombosis, AEDM: Acute disseminated encephalomyelitis, LETM: Longitudinal extensive transverse myelitis; BFP, Bifacial diplegia; SOVT, superior ophthalmic vein thrombosis (SOVT).

**Table 2.** COVID Vaccine analysis print UK (as of 27 August 2021).

| Neurological Complications | Pfizer-BioNTech BNT162b | Moderna mRNA-1273 | AstraZeneca Vaccine |
|---|---|---|---|
| Cerebrovascular accident | 322 | 7 | 1168 |
| Ischemic stroke | 33 | 1 | 140 |
| Hemorrhagic stroke | 9 | 0 | 39 |
| Guillain-Barré syndrome | 44 | 3 | 393 |
| Transverse myelitis | 26 | 0 | 81 |
| Bell's palsy | 402 | 32 | 551 |
| Cerebral venous sinus thrombosis | 28 | 1 | 205 |
| Optic neuritis | 24 | 3 | 51 |

**Table 3.** Vaccine adverse events reporting system (VAERS) results (as of 27 August 2021).

| Neurological Complications | Pfizer-BioNTech BNT162b | Moderna mRNA-1273 | Janssen |
|---|---|---|---|
| Cerebrovascular accident | 198 | 155 | 5 |
| Ischemic stroke | 175 | 114 | 47 |
| Hemorrhagic stroke | 59 | 34 | 14 |
| Guillain-Barré syndrome | 49 | 49 | 81 |
| Transverse myelitis | 87 | 63 | 25 |
| Bell's palsy | 1647 | 1322 | 194 |
| Cerebral venous sinus thrombosis | 25 | 22 | 29 |
| Optic neuritis | 40 | 39 | 8 |

## 5. Conclusions

We conducted a thorough literature search regarding post-COVID 19 neurological complications. We also presented a case series of three patients with vastly different neurological complications post-COVID 19 vaccination. The goal is to understand the importance that neuroimaging plays in the early detection, diagnosis, and subsequent treatment of severe conditions, including cerebrovascular accidents, thrombosis, and demyelinating inflammatory processes. Further studies and data are required to provide a substantial basis and formulation for guidelines on the early identification and neuroimaging study on post-COVID-19 vaccination-related neurological manifestation.

**Author Contributions:** Conceptualization, S.S. (Shitiz Sriwastava); data abstraction and data analysis, drafting the manuscript, A.K.S., S.H.K., M.A.C., D.N. and S.S. (Samiksha Srivastava); writing—original draft preparation, S.S. (Shitiz Sriwastava); writing—review and editing, S.S. (Shitiz Sriwastava) All authors have read and agreed to the published version of the manuscript.

**Funding:** This research received no external funding. The authors declare that the research was conducted in the absence of any commercial or financial relationships that could be construed as a potential conflict of interest.

**Institutional Review Board Statement:** IRB approval from West Virginia University with IRB protocol number 2012191877.

**Informed Consent Statement:** Patient informed written consent was obtained.

**Data Availability Statement:** Data was extracted from the articles published in PUBMED, Google Scholar. This will be provided on request.

**Conflicts of Interest:** The authors declare no conflict of interest.

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
