# Peer review of "Spectrum of Neuroimaging Findings in Post-COVID-19 Vaccination: A Case Series and Review of Literature"

_2035-8377, doi:10.3390/neurolint13040061_

Round 1

Reviewer 1 Report

Manuscript ID: neurolint-1424431

Type of manuscript: Review

Title:    Spectrum of Neuroimaging Findings in Post COVID-19 Vac- 2 cination a Case Series and Review of Literature

This is an interesting review article about  to review published literature regarding post-COVID-19 vaccination neurological complications and characterize neuroimaging findings from three case presentation.

 In spite of these appropriateness of topics, some careful considerations should be made.

  1. Methods

Add inclusion or exclusion criteria. Because I wonder why the authors include only “SARS-CoV-2”, “post Covid vaccination”, “neurological complica- 21 tions”, “Guillain-barre Syndrome”, “Transverse-myelitis”, “Cerebral Venous Sinus thrombosis” 22 and “Cerebral hemorrhage”.

  1. Methods

Why were the articles included from database inception beginning December 2020 ?

  1. In Table 1, can you find any specific characteristics according to post vaccination complications? Time duration? Treatment efficacy?

If not, it is not meaningful just to summarize the cases that have already been reported.

  1. In Table 2, the number of post vaccine complications is related to when the vaccine was introduced. If that is not taken into account, there is a risk of misinformation being provided to the reader.

Minor

  1. In figure 3 legend, which is correct? 3(B)?? or (3B)

  1. The reference part does not conform to the journal format.

Author Response

Ref. No.: Manuscript ID: neurolint-1424431

   Neurology International

Title:    Spectrum of Neuroimaging Findings in Post COVID-19 Vaccination a Case Series and Review of Literature

We would like to thank the reviewer for the time they invested in improving the quality of our manuscript. Please see attached point-by-point changes/corrections or explanations to the reviewers’ comments.

Reviewer 1:

Title:    Spectrum of Neuroimaging Findings in Post COVID-19 Vaccination a Case Series and Review of Literature

 This is an interesting review article about  to review published literature regarding post-COVID-19 vaccination neurological complications and characterize neuroimaging findings from three case presentation.

 In spite of these appropriateness of topics, some careful considerations should be made.

 R#1.1 Methods

Add inclusion or exclusion criteria. Because I wonder why the authors include only “SARS-CoV-2”, “post Covid vaccination”, “neurological complica- 21 tions”, “Guillain-barre Syndrome”, “Transverse-myelitis”, “Cerebral Venous Sinus thrombosis” 22 and “Cerebral hemorrhage”.

A#1.1 Thank you for your comments. We now have updated the inclusion and exclusion criteria.

Please see line #68-79, page #2

R#1.2 Methods

Why were the articles included from database inception beginning December 2020?

A#1.2 Thank you for your comments. We included the date as Dec 2020 because the approval dates of Pfizer and Moderna these were approved in Dec 2020.

 R#1.3 In Table 1, can you find any specific characteristics according to post vaccination complications? Time duration? Treatment efficacy?

If not, it is not meaningful just to summarize the cases that have already been reported.

A#1.4 Thank you for the comments. Table 1 the characteristic features were the most common PNS manifestation post vaccination were GBS where as the most common CNS manifestation were transverse myelitis.

R#1.4 In Table 2, the number of post vaccine complications is related to when the vaccine was introduced. If that is not taken into account, there is a risk of misinformation being provided to the reader.

 A#1.4 In Table 2, the number of cases considered are those that were reported to VAERS (Vaccine adverse reporting system, CDC US) and MHRA (Medicines and Healthcare products Regulatory Agency, UK) after the vaccination was introduced.

Minor

R#1.5 In figure 3 legend, which is correct? 3(B)?? or (3B)

      A#1.5 We now have corrected this 3(B).

      R#1.6 The reference part does not conform to the journal format.

      A#1.6 We have followed Vancouver reference style.

Reviewer 2 Report

The manuscript titled "Spectrum of Neuroimaging Findings in Post COVID-19 Vaccination a Case Series and Review of Literature" by Sriwastava et al is a nice study presenting an interesting case series with a thorough literature review documenting adverse neurological affects following COVID-19 vaccination. The findings of this study is significant especially during ongoing pandemic. The authors have presented all the studies a in nice manner, with a good coordination between methodology and discussion. The only thing I would advice is to add a strong future perspective section where authors should present their own original opinion on the usage of findings of this study in future research.

Author Response

Ref. No.: Manuscript ID: neurolint-1424431

   Neurology International

Title:    Spectrum of Neuroimaging Findings in Post COVID-19 Vaccination a Case Series and Review of Literature

We would like to thank the reviewer for the time they invested in improving the quality of our manuscript. Please see attached point-by-point changes/corrections or explanations to the reviewers’ comments.

Reviewer 2:

The manuscript titled "Spectrum of Neuroimaging Findings in Post COVID-19 Vaccination a Case Series and Review of Literature" by Sriwastava et al is a nice study presenting an interesting case series with a thorough literature review documenting adverse neurological affects following COVID-19 vaccination. The findings of this study is significant especially during ongoing pandemic. The authors have presented all the studies a in nice manner, with a good coordination between methodology and discussion. The only thing I would advice is to add a strong future perspective section where authors should present their own original opinion on the usage of findings of this study in future research.

A#2.1 Thank you for your positive feedback and considering our case for publication. We are

deeply pleased for the Editor’s and Reviewer’s guidance. We thank the reviewer for their time.

We now have added the opinion of the guidelines of future study in the conclusion.

Please see line #264-267, page #13

Reviewer 3 Report

The Authors report neurological features, MRI and CT scan findings and treatment outcomes of neurological complications following Covid 19 vaccination. They have reviewed the pertinent literature identifying 29 cases affecting central e peripheral nervous system,presenting also a cases series of three patients with neurological complications post vaccination. The authors did a good job, sound and useful in daily clinical practice, underlining how a  correct use of neuroimaging allows  an early diagnosis and adequate therapy.The article appears interesting and well documented worthy of publication on Neurology International.

Author Response

Ref. No.: Manuscript ID: neurolint-1424431

   Neurology International

Title:    Spectrum of Neuroimaging Findings in Post COVID-19 Vaccination a Case Series and Review of Literature

We would like to thank the reviewer for the time they invested in improving the quality of our manuscript. Please see attached point-by-point changes/corrections or explanations to the reviewers’ comments.

Reviewer 3:

The Authors report neurological features, MRI and CT scan findings and treatment outcomes of neurological complications following Covid 19 vaccination. They have reviewed the pertinent literature identifying 29 cases affecting central e peripheral nervous system,presenting also a cases series of three patients with neurological complications post vaccination. The authors did a good job, sound and useful in daily clinical practice, underlining how a  correct use of neuroimaging allows  an early diagnosis and adequate therapy.The article appears interesting and well documented worthy of publication on Neurology International.

A#3.1 Thank you for your positive feedback and considering our case for publication. We are

deeply pleased for the Editor’s and Reviewer’s guidance. We thank the reviewer for their time.